# Men’s Physical Stature: Tackling Heightism and Challenges in Fashion Consumption

**DOI:** 10.3390/bs12080270

**Published:** 2022-08-05

**Authors:** Osmud Rahman, Henry Delgado Navarro

**Affiliations:** Fashion at the Creative School, Toronto Metropolitan University, 350 Victoria Street, Toronto, ON M5B 2K3, Canada

**Keywords:** short male consumers, body image, fashion consumption, heightism, buying behaviour

## Abstract

In light of the limitations of previous research on fashion consumption by short men, the present study was undertaken to examine the relationship between male body image, height perceptions, clothing choice, garment fit, and heightism. In this study, 3D body scanning technology and in-depth interviews were employed to investigate the relational effects between men’s height, body image, and clothing consumption. In total, twelve men exhibiting a height of 5′ 8″ or less participated. The findings indicate that “heightism” is prevalent in today’s society. As well, the results reveal that overall appearance and physical stature become less important as people grow older. It is evident that short male consumers encounter challenges when they shop for fashionable and well-fitting clothing. They are underserved by the fashion industry and often impacted by judgmental biases. The study supports that garment alteration and buying clothes from the children’s section, or a bespoke tailor are not ideal solutions for short men. Furthermore, many of them just accept the fact that they are short and try to find ways to alleviate their frustration when consuming fashion.

## 1. Introduction

A substantial amount of academic research [1,2] has been undertaken to examine body image, appearance management, and clothing consumption. However, much of this prior research is primarily or exclusively focused on women. One plausible reason why little scholarly attention has been given to men’s body image and fashion consumption is due to greater sociocultural pressures on women’s appearance than on men. Another reason can be due to the false perception that men are less interested in grooming, physical appearance, and apparel shopping. By contrast, recent studies [3,4] have found that men are more concerned with their body image and physical appearance than generally assumed. According to Newman [5], between 1997 and 2009, male U.S. consumers doubled their spending in grooming products to US$ 4.6 billion. A more recent report released by Million Insights [6] also supports that the global market for male grooming products is projected to reach US$ 78.5 billion by 2025. Several studies [5,7,8] have proven that most men are concerned or dissatisfied with their bodies in general and their biceps, shoulders, chest, and overall muscle tone in particular. To enhance their physical appearance, men employ various appearance management strategies such as daily grooming, clothing and adornment practices, diets, high-impact workouts, the use of anabolic–androgenic steroids, and plastic surgeries to alter their body shapes and conform to the dominant bodily ideal norm. Although body fat, muscularity, and physical appearance can be altered through the aforementioned strategies, body height is difficult to alter without invasive intervention such as orthopaedic surgery (e.g., limb lengthening) or growth hormone therapy [9].

## 2. Context and Current State of Knowledge

A considerable amount of research has examined issues associated with apparel consumption behaviour. The taxonomy for the current study was developed from an existing body of work found in the literature on consumer behaviour, fashion studies, and body image. These include body image attitudes and behaviours among young men [5]; big and tall male apparel shoppers [10,11]; appearance management and clothing choice [12]; body types and apparel cues [13]; and garment fit and sizing [14]. The review of this rich and diverse extant research provides a primary focus for the current study and frames our later discussion and interpretation of fashion consumption by short men in the following areas: male body image, height perceptions, clothing choice, garment fit and sizing, and heightism.

Although some prior research has investigated male consumer behaviour, the specific topic of men’s height and clothing choice has received very limited attention. To the best of our knowledge, only two apparel studies [10,11] specifically focused on men’s height and shopping behaviour. However, these studies were conducted in the early 1990s and were limited to the “big and tall” segment of male consumers. To better understand the relationship between men’s physical stature, clothing choice, and garment fit, Ashdown’s [15] sizing system model also guided the present research. Ashdown’s model suggests that apparel sizing systems are linked to four key factors: the population measurements (body size and individual measurements), garment fit issues (fit quality), the design features (clothing style and construction), and communication of sizing and fit (apparel size labelling).

### 2.1. Men’s Physical Stature and Clothing Choice

Although physical stature plays a significant role in social perceptions and self-esteem, people do not always judge others based solely on their height. As discussed in previous research [16], the body can be decorated and altered through temporary, semi-permanent, and permanent modifications. Thus, height intersects with other physical attributes (e.g., body shape, muscle mass) and non-physical attributes (e.g., clothing, footwear) of appearance. In other words, body perceptions and self-satisfaction may vary among individuals depending on multiple factors. Thus, to understand the relationships between physical and non-physical attributes in social perceptions of men, it would be worthwhile and meaningful to examine the relationship between body height and clothing choice.

Clothing can be used as a non-verbal mediator between an individual’s self and their social environment. The way individuals dress can boost their self-esteem and confidence. In turn, clothes can influence other people’s perceptions of an individual. Frith and Gleeson [12] stated in their study that men who wanted to appear taller tend to use clothes that influences viewers’ perception of their height. For example, they may wear elevator shoes or garments that optically elongate their bodies. According to Swami and Harris [17], wearing a garment with vertical stripes gives the wearer a taller and slimmer appearance. Indeed, people often use clothing to manipulate how their body looks, conceal or camouflage their perceived physical faults, or accentuate desirable body parts. However, as Chattaraman et al. [18] assert, “although both men and women experience fit dissatisfaction, scholarly research has almost exclusively focused on women, leaving a critical gap in the research on men’s fit issues and preferences (p. 291)”. Thus, at present, there are still limited studies examining body image and garment fit for men. This gap in scholarship is even more pronounced with regard to short men.

According to a study [19] on body dysmorphic disorder, women are generally more dissatisfied with their bodies than men and more concerned with their weight and the appearance of their lower body (e.g., legs and buttocks). By contrast, men are more concerned with their muscle mass and upper body [19]. In addition, a considerable amount of research [13,20,21] on body image and apparel consumption indicates that many women use clothing to conceal their undesirable body parts. In other words, people who are less satisfied with their bodies (e.g., below average height, overweight) tend to choose clothes to manage the appearance of their bodies. Do these appearance-management strategies hold for short male consumers? Due to very limited empirical research focusing on this topic, it is important to gain an in-depth understanding of men’s height and body image in relation to clothing consumption.

### 2.2. Physical Stature and Garment Fit

Apparel fit defines how a garment conforms to the body [22] or the relationship between clothing dimensions and body dimensions [23]. Fit is frequently cited as the determinant factor in clothing evaluation and selection [14,24]. However, it is a challenge for many consumers to find a fashionable garment that fits them well in today’s ready-to-wear apparel market [25]. This is particularly the case for consumers whose body measurements are outside of the apparel standard sizing system, including short and overweight men.

According to Oliver et al. [8], one-third of men experience apparel fit problems resulting from issues with the length and circumference of shirt collars and waistbands and sleeve and pant lengths. Rahman and Yu [25] suggested that a well-fitting garment should look fashionable and be comfortable to wear. Another apparel study [16] found that male consumers with larger bodies preferred a looser fit for greater ease of movement and general comfort. Most importantly, these findings suggest that apparel fit is highly subjective, and a well-fitting garment should provide both physical comfort (e.g., ease of movement) and psychological comfort (e.g., aesthetically pleasing) to the wearers. Thus, when a garment fails to provide a comfortable fit, it leads to returns of merchandise, financial losses for retailers, as well as consumers’ dissatisfaction and psychological disturbance. For example, a study [8] reported that more than one-third of apparel returns are the result of poor fit. Hence, it is not difficult to understand why many consumers are frustrated with apparel shopping in general [26]. To advance our understanding of the relationship between men’s body image and clothing choice, this study investigated garment fit and consumer behaviour in relation to men’s physical stature.

## 3. Research Method

In this study, 3D body scanning technology and in-depth interview were used to investigate men’s body image and clothing consumption. This study was approved by the local Institutional Ethics Board. A C$ 20 gift certificate was offered to participants for compensating their time and travel expenses. All subjects provided their informed consent before taking part in the study. The ethics protocol of this study was approved by the Ethics Board of Toronto Metropolitan University, and the project identification code is 2018-376.

### 3.1. Subject Recruitment

In total, 12 male participants were recruited to participate in this study. The selection criteria of eligible participants were based on sex (male), age (18 years old and up), and height (5′ 8″ or less). All the participants were recruited from a large city in Canada. It is worth noting that 5′ 8″ and under was considered short for men’s height because this measurement has been used in previous research [11] and is used as a cut-off point to categorise average height of male consumers in North America.

### 3.2. Three-dimensional Body Scanning

To offer a more objective method and a comprehensive approach to assessing body image, a 3D body scanner was employed for collecting anthropometric data (accurate body measurements); these data were then used in conjunction with in-depth interviews for reinforcement and data triangulation. Among the twelve participants, only nine of them participated in the 3D body scanning session. Before the body scanning, a manual height-measuring device and a digital scale were used by the researchers to record participants’ height and weight. This allowed the researchers to gauge participants’ familiarity with their actual body measurements. For the body scan, the [TC]^2^ white-light 3D Body Scanner was employed. This digital equipment collects anthropometric data and generates a virtual 3D body image of each human subject. The [TC]^2^ software also extracts a series of extremely accurate body measurements. If a participant in our study expressed photo light sensitivity concerns on the consent form, he was recommended not to participate in this portion. To capture the most accurate measurements and 3D body image, participants undressed to their underwear in a private curtained cubicle area before entering the secluded scanning booth. The 3D body scanning session took about 10 min, followed by an in-depth interview.

### 3.3. In-Depth Interview

Through the in-depth interview, qualitative data was collected from the twelve participants. At this stage, the goal was to record their apparel shopping experiences, garment fitting issues, and challenges regarding choice when purchasing clothing. The one-on-one interviews allowed participants to share their experiences and viewpoints about body perceptions, fit, and clothing consumption without confining their responses to a limited number of choices. Based on the previous literature [16,21,23,27], a set of interview questions was developed aimed to illuminate specific thematic concerns encompassing: (1) short men’s body image; (2) clothing choice and apparel shopping experiences; (3) garment fit and body height; and (4) challenges, overall (dis)satisfaction, and possible solutions. Interview questions included “What challenges do short men encounter when they shop for clothing?”, “What are the relationships between body types and clothing fit?” and “Are there any specific appearance-management strategies that short men employ to enhance their outward appearance?” Each interview lasted between 35 and 55 min and was audio-recorded for subsequent transcription.

### 3.4. Data Analysis

The methods of content analysis and holistic interpretation were used for data examination. Content analysis is a systematic and objective technique to identify special characteristics relevant to a research topic. This method aims to synthesize common themes that emerge from interviews transcriptions. The content analysis procedure used in this study was inspired by Zimmer and Golden [28]. In this case, the interview transcripts were reviewed multiple times and the data was coded, organised, and analysed by two coders independently. Recurring attributes with similar characteristics were grouped into categories to facilitate further comparison. When discrepancies or disagreements occurred between the two coders, extensive discussion and reflection on the original texts were required to achieve a consensus and resolve the issues. Following Kassarjian’s [29] recommendations, Cohen’s kappa was used to calculate the intercoder reliability. The results indicate that the amount of agreement between two coders is good (kappa = 0.69) according to Fleiss [30].

## 4. Results

### 4.1. Scan-Derived Anthropometric Measurements

Based on the 3D body scanning data, the neck circumference of the sample ranged from 13.54″ to 16.24″, with a mean of 14.70″. In terms of sleeve length, the measurements of both sides were similar and ranged from 30.59″ to 34.87″, except for P1, P3, and P6, who exhibited a left sleeve length of about 1.5″ to 1.75″ longer than the right sleeve length. In terms of the girth measurements (circumference) at the chest, waist, and hips, the anthropometric data and 3D body images confirmed that the body sizes of P2, P3, P4, P5, and P8 were larger than the rest of the participants (see Figure 1). These findings are supported by the BMI classification of those participants. Moreover, among the nine participants, the length of the inseams was similar on both sides (less than 0.5″), ranging from 27.76″ to 30.59″. It is worth pointing out that, due to the focus and scope of this study, other anthropometric data points were not presented nor analysed in this study. In addition, it is important to note that subject P5 did not dress in the required tight-fitted underwear (e.g., briefs or trunks) for the 3D body scanning session. As a result, some measurements below the waistline of participant P5 are invalid; the 3D body scanner captured the measurements of the boxer briefs instead of the actual body dimensions.

### 4.2. Results of In-Depth Interview

As shown in Table 1, twelve men participated in the interview session, and the sample comprised diverse ethnicities. According to the results from a descriptive analysis, four participants earned between C$ 80,000 and 99,999 per year, and five of them hold a bachelor’s degree. In terms of the self-reported body information, none of the participants considered themselves as either tall, underweight, muscular, or obese, but four participants (P1, P4, P10, and P12) considered their height as medium or average. As shown in Table 2, the age of participants ranged from 21 to 72 years, and the mean age was 38.7. Each participant’s weight and height were measured without shoes to calculate their body mass index (BMI). The height for the sample ranged from 64.5 inches (5′ 4.5″ or 163.8 cm) to 68 inches (5′ 8″ or 172.7 cm) while the BMI ranged from 19.3 to 33.9 with an average of 24.26. Based on their BMI calculations, seven participants were considered as normal weight, four overweight, and one obese. Therefore, there are a few discrepancies between self-reported body information and actual body measurements. For example, three participants (P2, P3, and P8) would like to be considered or perceived as right weight or broad/thick boned instead of overweight or obese.

### 4.3. Age, Life Stages, and Height Perceptions

More than half of the participants (*n* = 7) were satisfied with their current body shapes (not specifically their height), one participant (P2) was dissatisfied, and four of them (P3, P4, P7, P8) expressed that there is room for improvement. During the interviews, many participants admitted that they felt the pressure to conform to the ideal body norms in North America (i.e., leaner, muscular) particularly when they were young. Now they tried to ignore the pressure or not pay too much attention to it. In terms of height, many of the participants just accepted the fact that their height is shorter than the average male in North America. Three participants also divulged that they have become comfortable with their height and seldom compare themselves to others. The following excerpts highlighted some of their feelings.

In terms of my height, I can’t do anything … I’m not going to walk around on stilts to conform to the ideal body norm … I try to ignore the perception of what other people think … I would say I do have the pressure, but I ignore it.(P11)

I’m used to being this height, and I don’t notice that I’m short until I see the photos. … sometimes you can tell from a wedding photo when you’re standing around with a bunch of people who are taller than you.(P8)

According to some participants’ responses (*n* = 5), they perceived height as more important when they were young due to self-esteem and dating concerns. Thus, it is reasonable to suggest that men’s age, life stages, physical stature, and dating potential are closely related. As Rahman and Yu [31] asserted in their study, “As people age and experience various life-changing events, many may change lifestyle choices, buying habits or develop new needs to fit their new assumed role. (p. 196)” The following two excerpts provide support for previous research on the shopping behaviour of aging consumers [31] and men’s body image at different life stages [32].

I think when you are in North America, there is a societal expectation of men’s height. … When I was young, I felt the pressure to beef up a little bit. … I think living as a short man, it’s difficult to find a suitable mate because tall women will never look at you. Only women who have the same height or shorter might look at you. … but now I am 39, and height is not important to me anymore because there is nothing I can do to get taller.(P5)

When you were young, you pay more attention to your body size and height. But now, I have a better understanding of myself and self-appreciation, and also when you’re surrounded by people you love, you don’t look for that to boost your self-confidence.(P10)

### 4.4. Body and Clothing Proportions

The study found that short men felt disappointed with their height not solely due to dating disadvantages but also due to its impact on their apparel shopping experiences. Moreover, the interview findings indicate that six participants were more concerned about their overall body proportions than their height when shopping for clothing. Furthermore, the interviews’ transcripts revealed that body proportions play a critical role in clothing selection because how the garment conforms to the human body is particularly important to self-presentation and identity formation. The following interview excerpts capture that:

The ideal body size for me is proportionate regardless if they’re tall, short, underweight, on weight, or overweight. … When buying clothes, I always consider my height. I’ve to look at the proportion of clothing to make sure that it makes sense on my shorter frame.(P1)

I knew some guys were a few inches taller than me, and they beefed up, and I remember looking at them and I was like, ‘You look ridiculous.’ I don’t want to be that bulky because it looked disproportionate to me.(P5)

However, it is important to note that several participants were more concerned about their body shape than height when shopping for clothing because altering a garment to fit an individual’s height seems easier to achieve than adjusting a garment to fit the body shape. In other words, lengthwise alteration of clothing to suit the body is less of a problem than widthwise alterations. For example:

If a garment doesn’t fit right at the chest or the waist, I wouldn’t buy it. Width is a little bit harder to tailor to fit the body size. Length can always be altered for the most part, so, the body shape is more important than the height.(P7)

Based on the above findings, it is reasonable to conclude that, in addition to physical stature, body shape plays a crucial role in clothing selection and consumption. Short men’s physiques are varied (mesomorph/natural muscular body type, ectomorph/lean body type, and endomorph/softer and rounded body type), and therefore, further research on the relationship between body height, weight, and shape seems to be imperative.

### 4.5. Clothing Fit and Style—Which Attribute Is More Important?

Prior research [33] indicated that fit and style are the two most important determinants of clothing choice. According to the interview transcripts, it seems evident that our study’s participants faced different challenges in finding well-fitting and stylish clothing. Some of them felt that they do not have many choices or had to make a trade-off decision between fit and style. For example, P11 said, “I’m willing to wear something bigger as a trade-off for better style”; another participant, P7, also said, “For the things [clothes] that fit my body, they tend to be more basic”. Furthermore, the following excerpts support these findings.

As soon as I step out of classic and try to go trendy, that’s where it becomes difficult. I find whatever’s on-trend right now is slim fit, it’s difficult for me because I’m thicker in the midsection. … Blazers and jackets are the biggest headaches for me. I might have to settle for a cut that I didn’t really want. … Honestly, now I’m lowering my expectations.(P5)

I would say it’s more difficult to find the right size than the style. … Although the style is more important to me, I’m willing to pay extra money to alter the style I really want.(P10)

### 4.6. Garment Fit and Alternation—Is Alternation a Solution?

Overwhelmingly, almost all the participants (*n* = 11) expressed that it has been a challenge for them to buy a well-fitting garment. From the following excerpts, it is indisputable that short men have been underserved by the fashion industry. Most clothing items do not fit their body types. In particular, the pants’ legs and sleeves are always too long and need to be hemmed or altered.

There are no pants that can fit lengthwise for me. They all get hemmed, but I think that I just accept the reality, and also in terms of the waist sizes, I find what works the best for me but it’s still too big.(P11)

If the pants are too long, I would buy them because they’re easy to be altered. I would not buy a shirt, jacket, or coat if the sleeve length is too long.(P10)

Interestingly, many participants expressed that clothing alteration is one of the remedies, a “quick fix” but not a perfect solution. It is not difficult to understand why many participants did not consider clothing alteration an ideal solution. This is due to several reasons: (1) it costs them extra money, (2) it is inconvenient and time-consuming, (3) some clothing styles are difficult/expensive to alter, and (4) individuals choose different ways to deal with the fitting issues (i.e., roll up the sleeves, tuck in the shirt, cuff the pants).

I know if I am buying pants, I will have to get them altered. It’s a pain. Do I want to pay for the alteration on top of the price of the clothing? As well as the time to drop it off for alteration and then go there to pick it up. Do I want that hassle?(P5)

Alternation is not a perfect solution. If you’re purchasing a pair of pants for $30 to $40 and then the alteration costs another $50, it doesn’t make sense.(P7)

If the sleeves were too long, I would still buy them if I liked the style. I’m the type of person who usually rolls his sleeves anyway.(P9)

I have to get everything hemmed or rolled up. I think I struggle the most in terms of buying clothes is the sleeve length.(P8)

### 4.7. Short Men Who Wear Children’s Clothing

Although many participants did not consider childrenswear as an alternative, three participants would not mind wearing children’s clothes. There are a few reasons why some short men still purchase apparel products from the children’s section. First, they can find clothing in better fit and style. Second, childrenswear costs less compared to menswear. Third, they do not need to spend extra money and time on alternation.

When I find clothes that are built for my measurements, sometimes they can only be found in the children’s section.(P9)

My shoe size is six and a half … so, very, very difficult … seven is one of those sizes that fly off the shelves. I tend to go and get the kid’s sizes and it’s cheaper.(P5)

However, shopping from the children’s section is not ideal for many adult male shoppers. Short men may not feel comfortable, or they may not find the clothing styles suitable for them. As one of the participants commented:

Although my friends do brag about being able to wear stuff in children’s size. The style might not be appropriate. Also, I would probably feel embarrassed waiting outside the fitting room with the kids.(P4)

### 4.8. Concealing the Fitting Problem

Other than garment alteration and buying from the children’s clothing aisle, hiding, or concealing undesirable garment parts is another fit-management strategy for short men.

It’s less difficult for me to buy a shirt because I can hide the fitting problem. I only wear certain shirts under a sweater. They look nice only showing the collar part, but they look like crap if it’s just the shirt. I buy them knowing that I will never, ever wear them if it’s not covered.(P12)

In terms of the heightwise, I find the tops are easier to fix because if it’s a little bit longer, you can tuck it in.(P11)

It is important to note that, although a considerable number of studies [14,24,33] have examined how clothing has been used to conceal or camouflage undesirable body parts, none of these studies were undertaken from the viewpoint of hiding and concealing undesirable garment parts due to poor fit. Thus, it would be interesting to further investigate the relationship between body image and garment concealment.

## 5. Conclusions—Implications and Recommendations

### 5.1. Body Types and Garment Fit

According to the findings, shopping for a stylish and well-fitting garment is a daunting task for short men. For example, all the participants in this study expressed that they often encountered challenges and difficulties when shopping for fashionable apparel to fit their bodies. Many participants felt that they were underserved by the fashion industry, and there is nothing that they could do to change the current sizing system. Many of them just accepted the fact that they are short and tried to find ways to alleviate their frustration. For example, the following phrases were recurrent throughout the interviews:“accept the fact that I am short”;“I get used to it” (the pants or sleeves are always too long);“I try to ignore the perception of other people” (about the interviewee’s body image);“I just accept the reality” (all the pants need to be hemmed);“I know how to circumvent that struggle” (about clothing fit);“I’m lowering my expectations” (about clothing style and fit)“It’s a sacrificial trade-off” (e.g., between style and fit);“I hope the garment will shrink”;“I just roll up the hem/sleeve”.

Based on the results, it is evident that garment alternation is not a perfect solution due to extra cost, additional time, and inconvenience. As one participant (P3) expressed during his interview, “I’m sure some people who are shorter than average height would like pants to fit them without alteration”. Likewise, as revealed by another participant, buying clothes from a children’s brand or a tailor is not an ideal alternative either. Many short men do not like to shop in the childrenswear section, finding the experience degrading. Similarly, not every consumer can afford to buy clothes from a bespoke tailor regularly. This is specifically the case for casual wear items such as jeans, t-shirts, hoodies, and jogger pants.

According to the interviews, many participants expressed that it has always been a challenge for them to find well-fitting garments because of the inconsistencies in the apparel sizing system. As P1 responded in the interview, “There is no standard, so 28 [referring to waist circumference size for pants] in one store will fit differently from a 28 in another store. Even within one company, when they change the style or the designer, the fit changes. … So shopping is difficult and fit becomes an issue as well”. Every brand and company use different size charts or specifications. For example, retailers Legendary Whitetails, Van Heusen, and Brookes Brothers use different measurement specifications for their sizing charts—there is no uniformity in terms of sizes (see Table 3).

When comparing the neck measurements from Figure 1 (anthropometric data) with those in Table 3 (men’s shirt size chart), all nine participants from the body scanning session can be classified as size “S” or “M” except P8. Although the neck measurements of P2, P4, and P5 fall into size “S”, the chest and waist measurements correspond to a size “L”. Therefore, it would be difficult for these participants to find a men’s shirt to altogether fit their neck, chest, and waist correctly. Given this observation, it is evident that many short men whose body mass places them in the overweight (P2, P4, P5) or obese category (P8) are more likely to encounter more clothing fit problems than men of short height who are lean or slim bodied. For example, P8 could wear a size “XL” dress shirt to fit his body shape but the sleeve length will be too long. To fix their fit and sizing issues, the global fashion industry would need to develop a universal and up-to-date sizing system, as well as offer a wider range of sizes and cuts/silhouettes (e.g., short/skinny, short/slim, short athletic, short/broad) to fit the diverse body types and satisfy different individuals’ needs and aspirations of male-identified consumers.

### 5.2. Height, Age, and Garment Fit

In most cases, older people already have a life partner. They are no longer seeking dating opportunities. Thus, it is reasonable to believe that physical appearance becomes less important to older individuals as compared to their younger counterparts. This finding underscores at least two important implications for our study. First, the salient effects and perceptions of height are associated with a person’s life stage. Secondly, research that employs multiple physical characteristics (weight, body shape, muscle tone) may yield different results from a study that merely focuses on a single attribute such as height. Thus, it is reasonable to suggest that body perception studies focus on a single attribute and may overemphasise or inflate the impacts of one specific physical characteristic. Therefore, additional research is needed to examine the relative importance of multiple physical attributes.

As people get older, some of their body characteristics may undergo the most obvious physical change, e.g., stooped posture. To accommodate these physical changes, age-specific adjustments need to be included in clothing sizing and apparel pattern drafting. For example, changing postures requires the extra length to be added in the upper part of the back of a coat or jacket. Other than adjusting or re-engineering individual garment parts to accommodate specific body or sizing needs, transformable apparel design may be another alternative [37]. If a garment can be transformed into different styles and silhouettes to fit different body types, consumers will be more likely to wear it for longer. In other words, transformable clothing can be used to fit different body types and satisfy diverse tastes (more versatile) as well as to extend the lifespan of a fashion product (making it more environmentally friendly) [38].

### 5.3. Heightism

In general, short men experience greater body dissatisfaction and frustrations than their taller counterparts, especially when looking for dating partners and well-fitting garments. Subtle forms of prejudice and unfair treatment against shorter men were recurrently emerging from the interviews. For example, several participants felt that they were underserved or ignored by the fashion industry. They faced many more challenges and difficulties in finding good-fitting and stylish clothing because many ready-to-wear fashion brands do not cater to them. Therefore, it is reasonable to suggest that “heightism” is rampant in today’s society and is evident in the fashion industry. As one participant (P1) expressed, “It’s weird that the short man syndrome hasn’t really translated in the fashion industry ... being big and tall or plus-sized is not a stigma anymore but being short still is. There aren’t stores that cater to short men”. In this interview excerpt, participant P1 provided another interesting observation about the current priorities of the fashion industry. Several Canadian stores cater specifically to “big and tall” men (e.g., George Richards Big & Tall Menswear and Mr. Big & Tall Menswear) and “plus-size” women (e.g., Penningtons and Addition Elle). However, there are no menswear stores that cater to short or petite men. One plausible explanation is that the phrases such as “small and short” or “short and petite” carry negative associative meanings (tiny, little), and due to hegemonic masculinity, many short men may not want to be labelled or perceived as short. In our society, height is often associated with positive and negative social perceptions and judgmental biases. For example, people may “look up” to someone who is successful and “look down” on someone who is perceived to be less capable, inferior, or less fortunate. As one may notice, both commonly used idioms express stature-based social bias. They imply that tall stature is associated with positive or desirable social characteristics, while short stature represents its opposite. The aforementioned is an example of heightism; a form of negative cultural bias that leads to prejudice and discrimination towards individuals and groups who exhibit a short stature while simultaneously rewarding tall individuals and groups. As Feldman [39] asserted, “… to be tall is to be good, and to be short is to be ‘stigmatised’ (p. 243)”.

## 6. Limitations and Future Research

Although the findings from this study provide meaningful insights and knowledge to academicians and fashion practitioners, future research on issues related to men’s physical stature still requires further investigation. Specifically, the current research has several limitations. First, the results cannot be generalised to short females or populations of other gender identities (LGBTQ2S+). Second, the current study is primarily focused on men’s height, but future research examining the relational effects of different physical attributes such as body shape, weight, height, BMI, and muscle tone is needed. Third, the current study’s sample size is relatively small compared to other qualitative research on men’s body image. Future research can benefit from a larger sample for improving generalization in other populations with different demographic profiles (e.g., women, seniors), physical statures (e.g., average height, tall), or body weights/types (e.g., overweight, underweight, slim, muscular). Fourth, height stereotypes of men could be different across cultures due to different socio-cultural values and ethnic-based phenotypical differences. Therefore, it would be interesting to expand this research into a cross-cultural longitudinal study to explore the differences in the perception of body image between different cultures over an extended time. Fourth, further study should be conducted to explore the relationship between naked body image and clothed body image because body and clothing are inexorably linked.

## Figures and Tables

**Figure 1 behavsci-12-00270-f001:**
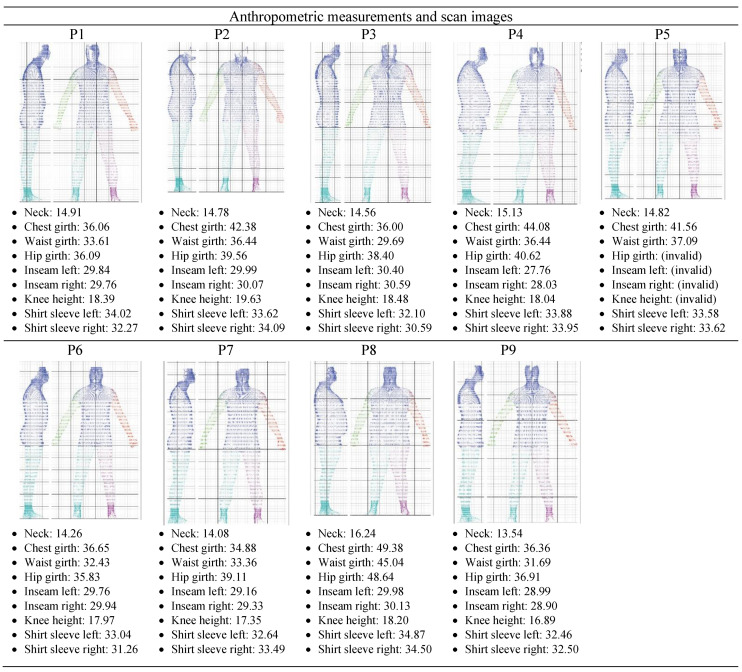
Scan-derived anthropometric measurements and body images of 9 participants.

**Table 1 behavsci-12-00270-t001:** Demographic and self-reprinted body information.

Demographic Information	Participants’ Codes (Age)
*Annual Income*	
Less than C$ 20,000	P9 (21)
C$ 20,000–39,999	P2 (24), P7 (29), P12 (72)
C$ 60,000–79,999	P3 (25), P5 (39)
C$ 80,000–99,999	P1 (48), P6 (43), P8 (35), P10 (50)
C$ 100,000 or above	P4 (55)
No response	P11 (23)
*Education*	
High School	P5
College/vocational school	P1, P8, P12
Bachelor’s degree	P2, P3, P7, P9, P11
Master’s degree	P4, P6, P10
*Ethnicity*	
Chinese	P3, P4
Latin American	P6
Caucasian	P8, P11, P12
Southeast Asian	P1, P2, P10
East Asian	P7, P9
Middle Easterner	P5
Self-reported Body Information	
*Height*	
Tall	P1
Medium	P4, P10, P12
Short	P2, P3, P5, P6, P7, P8, P9, P11
*Weight*	
Overweight	P4, P5, P8
Right Weight	P1, P2, P3, P6, P7, P9, P10, P11, P12
Underweight	None
*Body Type*	
Muscular	None
Fit/Athletic	P3, P6, P9, P10, P12
Thin/Skinny	P1, P7, P11
Broad/Thick Boned	P2, P4, P5, P8
Obese	None

**Table 2 behavsci-12-00270-t002:** Actual body information.

Actual Body Information	P1	P2	P3	P4	P5	P6	P7	P8	P9	P10	P11	P12	Total
*Age*	48	24	25	55	39	43	29	35	21	50	23	72	
*Height* (inches)	68.0	65.0	67.5	67.0	65.0	66.0	66.0	66.0	64.5	66.0	67.0	67.0	-
*Weight* (pounds)	135	162	163	170	160	130	135	210	134	154	123	140	-
*BMI*	20.5	27.0	25.1	26.6	26.6	21.0	21.8	33.9	22.6	24.9	19.3	21.9	-
*BMI Weight Status*													
Normal (18.5–24.9)	√	-	-	-	-	√	√	-	√	√	√	√	7
Overweight (25.0–29.9)	-	√	√	√	√	-	-	-	-	-	-	-	4
Obese (30.0 and above)	-	-	-	-	-	-	-	√	-	-	-	-	1

BMI Categories: underweight ≤ 18.5; normal weight = 18.5–24.9; overweight = 25–29.9; obesity = 30 or greater.

**Table 3 behavsci-12-00270-t003:** Men’s shirt size chart from three different fashion brands.

	Neck			Chest			Waist			Sleeve		
Size	LWT	VH	BB	LWT	VH	BB	LWT	VH	BB	LWT	VH	BB
S	14–14.5	14–15	15	34–36	34–38	37–38.5	28–30	28–32	31.5–32.5	32–33	32–33	-
M	15–15.5	15–15.5	15.5–16	38–40	38–42	39–41.5	32–34	32–36	33.5–35.5	33–34	32–33	-
L	16–16.5	16–16.5	16.5	42–44	42–46	42–44.5	36–38	36–40	36.5–38.5	34–35	32–35	-
XL	17–17.5	17–17.5	17–17.5	46–48	46–50	45–47.5	40–42	40–44	39.5–41.5	35–36	32–37	-

LWT: Legendary Whitetails [34]; VH: Van Heusen [35], BB: Brookes Brothers [36]

## Data Availability

Data sharing not applicable.

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
