# Peer review of "Men’s Physical Stature: Tackling Heightism and Challenges in Fashion Consumption"

_behavsci, 2022, doi:10.3390/bs12080270_

Round 1

Reviewer 1 Report

The authors address a very important issue that has been neglected in the fashion consumption scenarios but is also relevant for its body image and psychological impact of male customers who are ‘considered’ short according to their national/racial standards. The potential of the present work is a little diminished and should be emphasized in the title and abstract. They should be more compelled to reflect the powerful verbatim that is included in the results. Limitations are clearly identified and indicated. The verbatim is relevant to help fashion practitioners and academicians better understand this phenomenon's impact.In the next paragraphs, this reviewer will point to several aspects that still need the authors' attention to improve the Ms and make the most of their work.

The title should reflect the key problem ‘Heightism’ (maybe: “Men’s physical stature and fashion consumption: Challenges and preferences to tackle heightism.’

that can be more direct to the issue, informative and also help to raise awareness.

The rationale provided at the beginning of the introduction (lines 22-27) and the abstract (first sentence, line 8 and 9) are weak for the possibilities. These first lines/sentences should present ‘the problem' which is not just the ‘present study attempts to fill the void of a limited previous investigation’ but the authors should start with the targeted issue: much of the research on fashion consumption has been focused on women. And in men, Short Men’s Body Image and Clothing Choice has been neglected. There’s a heightism that is being explored here.
Also, for the introduction, please, do not start with the economic scenario but the opposite. I’d suggest starting the introduction with sentence 27 and moving the 3 first sentences to a later step, to indicate that it also has a big economic implication.

Please, discuss the limitations due to the sample size in recruitment (n=12) and ·D body scan (n=9)

Table 1 should be labeled as a figure and should be fitted to one page so the 9 patterns can be seen at once.

Table 2. I’d suggest doing a squared table with the left column as it is and a raw with P1 to 12, and then filling in the squares with colors, so a colored picture is more illustrative and easy to interpret, similar to Table 2.

Table 3. Please, use colors to provide a visual picture. I.e. underweight yellow, normal green, overweight orange, obesity red.

Thank you for your work

Author Response

The title should reflect the key problem ‘Heightism’ (maybe: “Men’s physical stature and fashion consumption: Challenges and preferences to tackle heightism’ that can be more direct to the issue, informative and also help to raise awareness.

The title has been changed to “Men’s Physical Stature: Tackling Challenges in Fashion Consumption and Heightism”

The rationale provided at the beginning of the introduction (lines 22-27) and the abstract (first sentence, line 8 and 9) are weak for the possibilities. These first lines/sentences should present ‘the problem' which is not just the ‘present study attempts to fill the void of a limited previous investigation’ but the authors should start with the targeted issue: much of the research on fashion consumption has been focused on women. And in men, Short Men’s Body Image and Clothing Choice has been neglected. There’s a heightism that is being explored here.
Also, for the introduction, please, do not start with the economic scenario but the opposite. I’d suggest starting the introduction with sentence 27 and moving the 3 first sentences to a later step, to indicate that it also has a big economic implication.

I have changed the first sentence of the abstract to strengthen and emphasise the research goals and motives. Likewise, the first few sentences of the “Introduction” section have been re-arranged for the same reason.

Please, discuss the limitations due to the sample size in recruitment (n=12) and ·D body scan (n=9)

The sample size of the current study is relatively small as compared to other qualitative research on men’s body image. Future research can utilize a larger sample for improving generalization in other populations with different demographics (e.g., women, children, seniors), physical statures (e.g., tall, average height), or body types (e.g., overweight, underweight, slim, muscular).

Table 1 should be labeled as a figure and should be fitted to one page so the 9 patterns can be seen at once.

We changed “Table 1” to “Figure 1” and also re-scaled the Figure to fit into one page.

Table 2. I’d suggest doing a squared table with the left column as it is and a raw with P1 to 12, and then filling in the squares with colors, so a colored picture is more illustrative and easy to interpret, similar to Table 2.

Thank you for your suggestion. I tried filling in colour on both Tables 2 and 3 but they don’t look consistent with Figure 1 and Table 4.

Table 3. Please, use colors to provide a visual picture. I.e. underweight yellow, normal green, overweight orange, obesity red.

Thank you for your suggestion. I tried filling in colour on both Tables 2 and 3 but they don’t look consistent with Figure 1 and Table 4.

Reviewer 2 Report

 It was a pleasure to review this paper and I believe that journal will receive a quality work, adequate for readers and suitable for future research. The structure of the paper is correctly done and easy to read, the literature is adequately cited and the research is correctly conducted.

It is recommended to make following changes and additions in the paper.

Limitation of the work is definitely a small number of examinee/sample. The title of the paper must be more specific and must better describe the topic of the work. Also, amend the abstract in such way that it describes the context to a lesser extent, and more the purpose and goal of research, as well as obtained results. I recommend authors to proofread English text. 

Author Response

The title has been changed to “Men’s Physical Stature: Tackling Challenges in Fashion Consumption and Heightism

I have changed the first sentence of the abstract to strengthen and emphasise the research goals and motives. Likewise, the first few sentences of the “Introduction” section have been re-arranged for the same reason.

The following information has been added in the "Limitation and Future Research" section:

"The sample size of the current study is relatively small as compared to other qualitative research on men’s body image. Future research can utilize a larger sample for improving generalization in other populations with different demographics (e.g., women, children, seniors), physical statures (e.g., tall, average height), or body types (e.g., overweight, underweight, slim, muscular)."

In addition, the revised manuscript has been copy-edited and proofread.

Reviewer 3 Report

In this study, 3D body scanning technology and in-depth interview were used to investigate men’s body image and clothing consumption. Research on the clothing consumption behavior of short men is an interesting topic on an important matter. Please see the following comments:

1) This manuscript discusses the clothing consumption problem of short men. The classification of consumer types based on height is novel and makes the paper interesting. Previous studies mainly focus on the clothing consumption of tall men, while this paper discusses the clothing consumption of short men from the opposite direction, which makes research in this area more comprehensive. It is important to note that even in short men there are large differences in physical stature. This paper introduces several variables to describe smaller men and a clearer description of the role of these variables should be included.

2) Another key issue that needs to be pointed out is that “2.1. Short Men's Body Image and Clothing Choice” did not properly focus on the concept of “Short Men's Body Image”. A more prominent discussion of research related to the physical characteristics of short men would be more consistent with your title.

3) Is there any writing error in “3.2.3. D Body Scanning”? Should it be “3.2. 3D Body Scanning”? If it is, please correct it. In addition, there is a lack of explanation for the purpose of this experiment regarding 3D Body Scanning.

4) In Table2, the demarcations of “Ethnicity” in “Demographic Information” are not clear. For example, the relationship between “Southeast Asian” and “Asian” is inclusive rather than parallel, I hope the author can change the way of demarcating “Ethnicity”.

5) There is a contradiction between the first sentence “More than half of the participants (n = 7) were satisfied with their current body shapes” in 4.3. and the first sentence “The study found that short men felt disappointed with their height not solely due to dating disadvantages but also due to their apparel shopping experiences” in 4.4., 4.3. stated that short men are generally recognized and satisfied with their height, while 4.4. stated that short men are generally dissatisfied with their height. Please be more precise in order to avoid semantic contradictions.

Author Response

1) This manuscript discusses the clothing consumption problem of short men. The classification of consumer types based on height is novel and makes the paper interesting. Previous studies mainly focus on the clothing consumption of tall men, while this paper discusses the clothing consumption of short men from the opposite direction, which makes research in this area more comprehensive. It is important to note that even in short men there are large differences in physical stature. This paper introduces several variables to describe smaller men and a clearer description of the role of these variables should be included.

Additional information has been added to Section 4.4. Please see the revised manuscript.

2) Another key issue that needs to be pointed out is that “2.1. Short Men's Body Image and Clothing Choice” did not properly focus on the concept of “Short Men's Body Image”. A more prominent discussion of research related to the physical characteristics of short men would be more consistent with your title.

In terms of consistency, we changed the sub-heading to “Men’s Physical Stature and Clothing Choice” to reflect the content as well as to match with the paper title.

3) Is there any writing error in “3.2.3. D Body Scanning”? Should it be “3.2. 3D Body Scanning”? If it is, please correct it. In addition, there is a lack of explanation for the purpose of this experiment regarding 3D Body Scanning.

It should be “3.2. 3D Body Scanning”. Also, an explanation has been added to this section.

4) In Table2, the demarcations of “Ethnicity” in “Demographic Information” are not clear. For example, the relationship between “Southeast Asian” and “Asian” is inclusive rather than parallel, I hope the author can change the way of demarcating “Ethnicity”.

It should be “East Asian” to be more precise.

5) There is a contradiction between the first sentence “More than half of the participants (n = 7) were satisfied with their current body shapes” in 4.3. and the first sentence “The study found that short men felt disappointed with their height not solely due to dating disadvantages but also due to their apparel shopping experiences” in 4.4., 4.3. stated that short men are generally recognized and satisfied with their height, while 4.4. stated that short men are generally dissatisfied with their height. Please be more precise in order to avoid semantic contradictions.

To avoid any misunderstanding, we added “(not specifically about their height)”. Participants had different responses towards their body shape and body height.

“More than half of the participants (n = 7) were satisfied with their current body shapes (not specifically about their height), one participant (P2) was dissatisfied, and four of them (P3, P4, P7, P8) expressed that there is room for improvement. … In terms of height, many of them just accepted the fact that they are shorter than the average height of males.”